# Neoadjuvant Therapy for Primary Resectable Retroperitoneal Sarcomas—Looking Forward

**DOI:** 10.3390/cancers14071831

**Published:** 2022-04-05

**Authors:** Alexandra C. Istl, Alessandro Gronchi

**Affiliations:** 1Division of Surgical Oncology, Johns Hopkins Hospital, Baltimore, MD 21287, USA; aistl1@jh.edu; 2Department of Surgery, Fondazione IRCCS Istituto Nazionale dei Tumori, Via Venezian 1, 20133 Milan, Italy

**Keywords:** retroperitoneal sarcoma, neoadjuvant, radiation therapy, chemotherapy, STRASS

## Abstract

**Simple Summary:**

This review summarizes the current evidence around neoadjuvant radiotherapy and systemic therapy for retroperitoneal sarcoma (RPS). While surgery is the cornerstone of treatment for these tumors, data from prospective studies, retrospective studies, early phase trials, and—most recently—our first phase III randomized trial for RPS suggest there are clinical scenarios in which neoadjuvant therapy may provide benefit. This review evaluates the STRASS results in the setting of other recent studies, identifies active trials of interest, and suggests future directions of study in this field. The intersection of STRASS and STRASS2 is considered and a summary of current acceptable approaches to neoadjuvant therapy for RPS is provided.

**Abstract:**

The cornerstone of therapy for primary retroperitoneal sarcomas (RPS) is complete surgical resection, best achieved by resecting the tumor en bloc with adherent structures even if not overtly infiltrated. Until recently, trials designed to elucidate the role of neoadjuvant radiation or chemotherapy for RPS have been unable to achieve sufficient enrollment. The completion of the STRASS trial, which explored neoadjuvant radiotherapy for primary resectable RPS, is a major milestone in RPS research, but has prompted further questions about histology-driven treatment paradigms for RPS. Though it was ultimately a negative trial with respect to its primary endpoint of abdominal recurrence-free survival, STRASS produced a signal that suggested improved abdominal recurrence-free survival with neoadjuvant radiotherapy (RT) for patients with liposarcoma (LPS). No effect was seen for leiomyosarcoma (LMS) or high-grade dedifferentiated (DD) LPS, consistent with recent literature suggesting LMS and high-grade DD-LPS have a predominant pattern of distant rather than local failure. These results, along with those from other recent studies conducted at the bench and the bedside, emphasize the importance of a histology-specific approach to RPS research. Recent evidence for patterns of distant failure in LMS and high-grade DD-LPS has prompted the initiation of STRASS2, a study of neoadjuvant chemotherapy for these histologies. As this study unfolds, evidence may emerge for novel systemic therapy options in specific sarcoma histotypes given the explosion in targeted and immunotherapeutic applications over the last decade. This article reviews current and recent evidence around neoadjuvant radiation and chemotherapy as well as avenues for future study to optimize these treatment approaches.

## 1. Background

Despite the rarity of soft tissue sarcomas (STS), we have made substantial strides in the study and management of these tumors in recent years. Gastrointestinal and translocation-driven sarcomas have benefitted enormously from the discovery of molecular targets and advances in targeted therapy; we are identifying histology-specific differences in sarcoma tumor microenvironments that may allow immunotherapy to change the landscape of sarcoma as it has with other malignancies; and—for the first time—we have made the successful execution of phase III trials for retroperitoneal sarcomas (RPS) a reality. Ultimately, the biggest effector of change in global sarcoma care over the last several decades has been a concerted effort toward international collaboration.

While trials that are currently recruiting and on the horizon will explore various facets of neoadjuvant therapy, the majority of our evidence for RPS is still retrospective. Though collaboration has made the products of our retrospective data collection far more valuable, the heterogeneity of these tumors still makes it challenging to design prospective studies that are both meaningful and pragmatic. With no high-level evidence definitively favoring a specific approach to neoadjuvant therapy for RPS, the use of radiotherapy and chemotherapy in this setting remains variable across countries and institutions. This review discusses the current literature surrounding neoadjuvant radiotherapy and systemic therapy for RPS, as well as active trials and future directions.

## 2. Evidence Surrounding Neoadjuvant Radiotherapy

### 2.1. Historical Context

In the retroperitoneum, STS in general have a propensity for local recurrence (LR) rather than metastatic spread [1,2]. A complete surgical resection remains the therapeutic goal for localized RPS; however, the size and extent of these tumors often make margin-negative (R0) resection impossible [3,4]. Many pathologists no longer evaluate margins for RPS and it is commonly accepted that ‘R0’ resections are the result of insufficient sampling. Large observational studies have reported anywhere from 34% to 69% of patients having a microscopic margin-positive (R1) resection [2,4,5,6,7], which significantly increases local recurrence risk and is associated with worse overall survival [5,6,8,9]. Comprehensive database studies have described predictors for incomplete resection, as well as the impact of an incomplete resection on survival, but have not specifically examined the impact of neoadjuvant radiotherapy (RT) on outcomes in these patients [6,9].

In the setting of high rates of local failure, multimodality locoregional therapy may have considerable value for some patients with resectable disease [10]. However, until recently, high-level evidence for the use of RT in STS was limited to studies of the extremities [3,11]. Before STRASS, the only phase III trial designed to explore neoadjuvant radiation for RPS met with poor accrual [12]. This may be due to the unique challenges of exploring RT for RPS: first, providers hesitate to delay curative surgery (or enroll patients in a trial where curative surgery could be delayed) in the absence of strong evidence to support neoadjuvant RT. Second, there is often hesitancy to administer retroperitoneal radiation given significant toxicity to surrounding structures. Results from early-phase prospective trials at high-volume institutions have demonstrated the feasibility of a neoadjuvant RT approach [13,14,15,16]. However, toxicity to proximate structures with unclear survival or recurrence benefit remains a concern. Recent systematic reviews have reported improved overall survival (OS) and LR rates across cohort studies exploring the benefit of RT for RPS; however, they have also demonstrated evidence of increased toxicity over surgery alone [3,17]. In the setting of recent results from STRASS—our first phase III randomized controlled trial for RPS [18]—optimizing patient selection for neoadjuvant RT and mitigating its toxicity will be fundamental to reframing our neoadjuvant treatment paradigms.

### 2.2. Histology-Specific Evidence

While STRASS was ongoing, the Transatlantic Retroperitoneal Sarcoma Working Group (TARPSWG) published a study of over 1000 RPS that demonstrated a significantly higher rate of distant failure for patients with LMS, who also had a relatively low rate of local recurrence compared with other histologic subtypes [19]. A significant association with worse progression-free survival (PFS) and distant failure rather than local recurrence for LMS compared with LPS was similarly noted in a phase I/II trial of intensity-modulated radiotherapy (IMRT) and intraoperative radiotherapy (IORT) for RPS [16]. The tendency of LMS toward distant recurrence may be particularly pronounced in primary retroperitoneal tumors: in a study of 115 LMS patients (of whom 47 were RPS), those with retroperitoneal primaries had a significantly higher risk of distant failure after surgical resection compared with other primary sites [20]. For LMS patients undergoing a wide excision—our standard of care for RPS—local recurrence-free survival (RFS) at 5 years was approximately 80% [20]. This was not stratified by preoperative radiotherapy receipt, but given the predominant pattern of distant rather than local failure for LMS, we might anticipate that RT would provide little additional benefit over surgery alone.

With respect to radiosensitivity, there is no in vitro or clinical evidence available that suggests LMS is a radiosensitive histology. One in vitro study exploring multiple LMS cell lines reported considerable heterogeneity in response to radiotherapy across cell lines, with only a uterine LMS cell line demonstrating poor survival after exposure to radiation [21]. In contrast, LPS radiosensitivity has been explored in a number of studies (although less commonly in the retroperitoneum). Though clinically and pathologically distinct from the well-differentiated (WD) and de-differentiated (DD) LPS of the retroperitoneum, myxoid LPS has demonstrated a consistent response to radiotherapy in a number of studies [21,22,23]. Unfortunately, the same radiosensitivity has not been definitively demonstrated for WD/DD-LPS, but the local-recurrence benefit of RT for retroperitoneal LPS suggested by the STRASS trial is discussed below.

Irrespective of histotype, genetic signatures may contribute to RPS radiosensitivity. This was explored in a cross-validation study of radiosensitivity with a 26-gene panel in sarcoma patients who had variable responses to radiotherapy [24]. While distinct tumor characteristics such as primary site may modify the utility of gene signatures in predicting response to radiation, these tools may be valuable in developing a personalized approach to neoadjuvant therapy for RPS. TARPSWG has initiated an international, prospective observational study of RPS that will enable histology-specific analyses in surgical specimens [25]. The Retroperitoneal Sarcoma Registry (RESAR) will be collecting these data—among many others—with the ultimate goal of improving the understanding of the natural history of this family of tumors, identifying new biomarkers that may be used to explore histology-specific responses to neoadjuvant chemo- or radiotherapy and to better predict oncologic outcomes.

### 2.3. STRASS-1

The STRASS trial investigating neoadjuvant external beam radiotherapy (EBRT) for primary localized resectable RPS enrolled 266 patients across 31 international centers (Table 1) [18]. It was a negative trial based on the primary endpoint of abdominal recurrence-free survival (ARFS), where failure was defined as any of: progressive disease during RT (local or distant), transition to inoperability, peritoneal disease at surgery, an R2 resection, or local recurrence after resection. This composite endpoint was developed to account for factors that might render patients unresectable while on RT, as well as to ensure a statistical sample size calculation that would permit sufficient accrual.

During STRASS’ development, histology-specific data that may have informed its design were not available, and so it was not powered to detect histologic differences in outcomes. This has ultimately led to some variance in data interpretation. There was a signal for improved ARFS in LPS patients after a sensitivity analysis whereby local progression or becoming unfit for surgery during RT were not counted towards the primary outcome as long as patients were eventually operated on. The justification of this sensitivity analysis is important to consider: the idea that pre-operative local progression is acceptable as long as it does not translate into either unresectability or an R2 resection is reasonable. Furthermore, the three patients who developed distant metastases while on RT were probably spared a morbid and oncologically ineffective operation. However, while 15 of the 19 patients who progressed while on radiotherapy still had a macroscopically complete resection, it is important to consider the possible added morbidity of a larger operation, even if a macroscopically complete resection is still achievable. Additionally, the time period between enrollment and surgery for the radiotherapy group (up to 21 weeks within the planned protocol) is substantial, and—in a post-STRASS world—may be a deterrent for both patients and providers considering neoadjuvant RT. Finally, irrespective of progression, if severe radiation-associated toxicities are precluding patients from surgery, we must think more carefully about the true benefit of neoadjuvant RT (or at least, the method in which it is currently delivered). Thirty-nine percent of the RT group in STRASS had at least a grade III complication over the course of the study. While the grade III+ complication rate in this group was only 14% pre-operatively, it is difficult to quantify the impact of radiation on post-operative morbidity.

One source of controversy surrounding the STRASS trial was the variability in the initial eligibility assessment and enrollment across centers. This was revealed in the setting of the STREXIT study, an analysis of all patients who were screened using STRASS eligibility criteria at the ten top recruiting centers [26]. STREXIT identified all patients with primary RPS at these ten participating centers who were not enrolled in STRASS during the same period and found that 57% of these (N = 473) were eligible for enrollment. Non-enrollment in STRASS was due to provider preference for 209 patients. A retrospective analysis of these patients stratified by receipt of neoadjuvant RT demonstrated a similar but non-significant trend toward improved ARFS in the LPS subgroup. When pooled with the STRASS cohort however, the WD-LPS and grade I–II DD-LPS group had significantly improved ARFS with RT. The pooled LMS and grade III DD-LPS groups demonstrated no significant difference in ARFS [26]. Ultimately, these studies exhibit the same trends observed in the subgroup analysis of STRASS, where no definitive evidence to support neoadjuvant RT for LMS or high-grade tumors was shown. At follow-up of the overall study cohort to date, there is no impact of neoadjuvant RT on OS.

With mixed interpretations of this trial, the approach to neoadjuvant RT for RPS remains variable across institutions: centers that did not routinely use RT in the neoadjuvant setting feel their approach has been validated, while centers that favor RT continue to offer it. Some institutions that previously administered neoadjuvant RT selectively have stopped offering it to LMS patients, while considering it more consistently for WD and grade I–II DD-LPS patients. However, the same controversies regarding the true merit of neoadjuvant RT, taking into account delays to definitive surgery and significant rates of toxicity, remain.

We can expect an update from STRASS in three years, which may provide more insight into late adverse events and persistent trends in survival outcomes. In the meantime, early-phase trials and other studies designed to explore the trends reported in STRASS should focus on the issues that have limited the uptake and translation of our results to date: the necessity of histology- and site-specific studies, reducing the duration of neoadjuvant therapy and subsequent delays to surgery, and mitigating or circumventing the toxicity associated with standard-course photon-based radiotherapy.

### 2.4. Short-Course RT

The rate of tumor progression in the STRASS study, which allowed up to 21 weeks to complete standard-fractionation RT and proceed to surgery, was 16%. While most of these patients still underwent a complete resection, potential increases in post-operative morbidity and the extent of surgery related to this progression remain a concern. The incidence of progression may be mitigated by a hypofractionated approach to radiation, which delivers a greater total dose to the clinical tumor volume in fewer treatments and may get patients to surgery several weeks earlier. The benefit of this approach was recently explored in a study of neoadjuvant short-course RT in extremity and trunk sarcomas [27]. Patients completed their RT and surgical resection in a median 20 days, had a post-operative adverse event rate of 18%, and experienced negligible RT toxicity [27].

While there are different anatomic considerations in the retroperitoneum, the shift from standard-fractionation EBRT to short-course or hypofractionated EBRT has met with successful outcomes in rectal cancer. Multiple phase III randomized trials have demonstrated equal efficacy and reduced toxicity with short-course RT [28,29]. This data exemplifies how short-course or hypofractionated neoadjuvant approaches may avoid the delays to definitive surgery that have limited the enthusiasm for preoperative radiation in resectable RPS. Further studies may help us understand whether a hypofractionated preoperative RT approach for RPS is equal in efficacy to conventional EBRT.

### 2.5. Non-Photon RT

During preoperative radiation, 14% of patients in the STRASS trial experienced a serious adverse event (grade III+), with 39% having a serious adverse event during study follow-up, and one death related to RT complications [18]. However, these high toxicity rates might be mitigated with newer RT modalities such as proton beam therapy (PBT). Target coverage is comparable between PBT and conventional IMRT, but a significantly lower integral dose can be observed with proton beams [30]. Proton beams drop off quickly beyond their targeted depth resulting in a negligible exit dose and thereby reduce toxicity to surrounding structures. Reported overall toxicity rates for PBT are consistently 14–18%, with minimal serious adverse events across multiple studies [31,32,33,34,35]. This technology also allows us to administer highly targeted radiation doses to margins at risk, where critical structures would otherwise preclude appropriate doses of photon-based radiation from being applied. One phase I trial of 28 fractions of PBT in RPS with a boost to the high-risk margin resulted in no dose-limiting toxicities [31].

FLASH PBT radiotherapy techniques have also been explored in preclinical studies with interesting results. FLASH RT is a form of radiation that uses electrons or protons delivered at high dose rates (>40 Gy per second) for short radiation durations [36,37]. Capitalizing on physiologic differences between tumor tissue and healthy tissue, the benefit of FLASH RT is its reduced toxicity to normal surrounding structures [36,38]. In particular, in vivo murine studies of FLASH RT have demonstrated reduced gastrointestinal radiation-associated toxicity compared with standard proton or electron beam radiation [39,40]. This finding is especially pertinent to RPS, where bowel in close proximity to the tumor is particularly susceptible to radiotoxicity. Considering the theoretic benefit of such a radiation modality for sarcoma, a recent study of FLASH PBT for sarcomas in murine and canine models demonstrated equivalent tumor control with reduced collateral tissue damage compared with standard PBT in mice [37]. Ultimately, while there are inherent challenges in translating FLASH RT to the clinical realm for deep tumors such as RPS, proton-based therapy may certainly aid in this transition.

The applications of PBT will only grow over the next several years. As histology-specific sarcoma trials exploring neoadjuvant RT develop, we should acknowledge PBT as an equally effective and far safer radiation modality for the treatment of RPS. Even if the value of neoadjuvant RT for radiosensitive RPS histologies can be demonstrated, safer and more effective methods of administering this radiation will be integral to the acceptance of RT as part of a neoadjuvant paradigm.

## 3. Evidence Surrounding Neoadjuvant Systemic Therapy

### 3.1. Historical Context

While cytotoxic chemotherapy is considered standard of care for patients presenting with metastatic sarcoma, there is no high-level evidence that systemic therapy in the neoadjuvant setting confers any benefit with respect to local recurrence, distant recurrence, or overall survival for patients with resectable RPS. Trials aiming to explore neoadjuvant or adjuvant chemotherapy for STS have commonly excluded RPS or included a minority of RPS patients [41,42,43,44,45], and no trials have specifically assessed neoadjuvant therapy for RPS.

In the absence of compelling evidence that chemotherapy improves oncologic outcomes for RPS, what are the goals of neoadjuvant chemotherapy for these tumors? First, neoadjuvant chemotherapy treats clinically undetectable micrometastatic disease, which may translate into reduced distant recurrence rates or longer disease-free intervals. It allows us to test the patient’s clinical response to standard systemic therapy regimens, which may inform treatment decisions in the event of recurrence or metastasis, as well as their histopathologic response, which has been found to correlate with survival and recurrence [46,47]. It enables us to select out patients with unfavorable disease biology and avoid morbid operations with no oncologic benefit in patients who progress on chemotherapy. Finally, tumor response to neoadjuvant therapy may allow us to perform both a safer and less radical resection. While the surgical standard for RPS is a complete surgical resection and this approach should not be altered based on an anticipated pathologic response to chemotherapy, response to chemotherapy—especially for histologies with lower rates of local recurrence—may allow us to be judicious in our decisions to resect less intimately involved viscera that are especially prone to complications (e.g., pancreas and duodenum).

At this time, RPS guidelines from the European Society for Medical Oncology (ESMO), European Reference Network for rare adult solid cancers (EURACAN), and Trans-Atlantic RPS Working Group (TARPSWG) reiterate the lack of evidence for neoadjuvant cytotoxic chemotherapy in resectable RPS, with caveats for a select few histologies that are discussed below [48,49]. When neoadjuvant chemotherapy is considered for tumors that are especially high risk, or where upfront resection will be difficult, systemic treatment should mirror first-line therapy for unresectable disease with an anthracycline-based regimen.

Outside the traditional cytotoxic chemotherapy paradigm, applications of targeted therapy and immunotherapy in other cancers are expanding. With a growing understanding of the heterogeneous immune signatures across sarcomas and encouraging findings from in vitro studies exploring common RPS gene expression profiles, the role for these therapies in RPS is likely to expand significantly in the coming years. This section reviews existing literature and active trials for cytotoxic therapy, targeted therapy, and immunotherapy in RPS.

### 3.2. Cytotoxic Chemotherapy

When chemotherapy is indicated for RPS, most commonly in the unresectable or metastatic setting, standard first-line therapy is an anthracycline-based regimen (doxorubicin or epirubicin), often with ifosfamide +/− mesna. Doxorubicin + dacarbazine is commonly used in LMS. Single-agent doxorubicin or gemcitabine + docetaxel are other acceptable regimens, although gemcitabine and docetaxel are not traditionally recommended as first-line therapy since this combination has worse toxicity than doxorubicin alone with no added benefit [50]. Other second-line agents such as trabectedin (an alkylating agent), eribulin (a microtubule inhibitor), and pazopanib (a tyrosine kinase inhibitor that targets vascular endothelial growth factor receptor (VEGFR) and platelet-derived growth factor receptor (PDGFR)) are considered inferior to anthracycline-based regimens. However, they have shown some histology-specific efficacy and could therefore be considered in the preoperative setting for high-risk, technically unresectable patients with contraindications to first-line therapy.

Three previous systematic reviews have evaluated the role of chemotherapy in treating STS, with the most recent including cohort studies as well as randomized trials [51,52,53,54]. However, most studies included in these reviews evaluated many STS sites or histologies, and the heterogeneity across studies was consequently so great that—at best—these reviews serve as a call to action for site-specific and histology-specific trials.

Since no trials to date have evaluated systemic therapy in the neoadjuvant setting for RPS alone, any prospective evidence we have is extrapolated from extremity and trunk populations, which have different patterns and mechanisms of local recurrence and distant spread. One European Organization for Research and Treatment of Cancer (EORTC) phase II trial explored neoadjuvant chemotherapy for high-risk adult STS excluding RPS. They defined ‘high-risk’ as ≥8 cm, locally recurrent, or grade II/III [43]. Their population of 134 patients included many STS histologies, and the extent of surgery was variable. Furthermore, patients were recruited over a long period of time, and doses of anthracycline and ifosfamide were lower than those considered active today. Ultimately, the trial was not powered to detect any difference in outcome between the study arms, and they showed no difference in median OS, with effectively equivalent OS and RFS at 5 years. Time to relapse was slightly but insignificantly longer in the neoadjuvant population. In the setting of their limitations, it is unclear how these data for extremity STS would meaningfully translate to the retroperitoneum.

Evidence specific to RPS for neoadjuvant chemotherapy is sparse and has been drawn from retrospective cohort and large database studies. A National Cancer Database (NCDB) study of RPS patients who received adjuvant or neoadjuvant chemotherapy reported significantly worse survival with chemotherapy [55]. However, the majority of patients received adjuvant rather than neoadjuvant chemotherapy, which seemed to be selectively administered to patients with higher risk tumors or R2 resections [55]. One retrospective review reported that 21.2% of RPS patients resected at their institution had received neoadjuvant chemotherapy, of whom 57.7% were high grade [56]. While they did not report specific survival outcomes for patients treated versus not treated with neoadjuvant chemotherapy, receipt of chemotherapy was not associated with a difference in OS or PFS. Rates of progression while on therapy were variable and were not reported in the context of grade or histology [56]. Similarly, a multi-institutional cohort study of 1007 RPS did not find that chemotherapy significantly impacted OS or rates of local or distant failure [19]. However, it is important to recognize that the influence of chemotherapy on survival differs based on pathologic response. One cohort of 55 RPS patients who received neoadjuvant chemotherapy and preoperative radiation were followed to evaluate survival in relation to tumor response [47]. Five year disease-specific survival (DSS) was 47% for all-comers but 83% for pathologic responders (25% of the cohort) and 34% for non-responders (*p* < 0.01) [47]. Pathologic response was associated with OS on multivariate analysis. Further research is required to identify the population of responders who will benefit from neoadjuvant chemotherapy *a priori*.

### 3.3. Histology-Specific Evidence

Efforts to narrow the scope of STS studies and collect histology-specific data have not only improved our understanding of histology-specific tumor properties, such as chemo- and radiosensitivity, but have reinforced the importance of such an approach. While there has been no demonstrated benefit of neoadjuvant or adjuvant chemotherapy for RPS to date, most of our data in STS is from all-comers and does not take the variable chemosensitivity across histologies into account. LMS in the retroperitoneum is considered moderately chemosensitive, while chemosensitivity for LPS varies based on differentiation [57,58]. It is accepted that WD-LPS is relatively chemoresistant, while DD-LPS or pleomorphic LPS have chemosensitivity to specific agents [57,58]. Synovial sarcomas and LMS are the only histologies named in RPS guidelines that are considered acceptable cases for consideration of neoadjuvant chemotherapy based on their chemosensitivity, even when amenable to upfront resection. Some retrospective studies from high volume centers have failed to consistently show improved oncologic outcomes with chemotherapy for synovial sarcoma [59,60]; however, others have demonstrated benefit for high grade or large tumors [61,62,63]. In the rare instances when synovial sarcoma arises in the retroperitoneum, there is a preoperative synovial sarcoma-specific nomogram that may help guide decision-making in the neoadjuvant setting [64].

Myxoid or round-cell liposarcomas have shown response to trabectedin in a phase III study; however, these results should be interpreted with caution in reference to RPS; myxoid LPS do not occur in the retroperitoneum, and if identified in a RPS biopsy, pathology review should be undertaken and investigation for a primary extremity tumor should be initiated.

The proclivity of LMS for distant metastasis has allowed for both retrospective studies and clinical trials analyzing histology-specific outcomes in advanced or metastatic LMS. For anthracycline-based combination therapy, a recent retrospective study of advanced LMS from contributing EORTC institutions found that first-line therapy with doxorubicin + dacarbazine improved median OS (36.8 months) compared with either doxorubicin + ifosfamide (21.9 months) or doxorubicin alone (30.3 months) in a propensity-matched cohort [65]. This effect was most pronounced after 18 months. A randomized phase II trial of gemcitabine with and without docetaxel for advanced LMS patients who had failed an anthracycline-based regimen found both gemcitabine-based regimens to be effective as second-line therapy [66]. Worse PFS was found for combination therapy compared to gemcitabine alone. Gemcitabine + docetaxel has also been tested as a first-line regimen in the unresectable setting, [67], but its reported median OS still fell short of that reported for both the doxorubicin monotherapy and doxorubicin + ifosfamide arms of a phase III study for advanced STS [68] and ultimately had worse outcomes than doxorubicin in a head-to-head comparison [50]. As such, in the unresectable setting, anthracycline-based therapy for LMS remains the standard of care.

In the neoadjuvant setting, gemcitabine + dacarbazine was tested against epirubicin + ifosfamide for extremity or superficial trunk LMS in a phase III randomized trial of standard chemotherapy versus histotype-tailored chemotherapy [69]. For LMS and other common histologic subtypes, no additional benefit was derived from a histology-tailored approach to chemotherapy [69]. However, TARPSWG conducted a retrospective analysis of RPS patients who received neoadjuvant therapy, and a subgroup analysis of LMS patients who received doxorubicin + dacarbazine (rather than the gemcitabine-based regimen in the histotype-tailored trial) had, at least, a slightly higher proportion of partial responses than LMS patients who received other regimens [70].

Ultimately, findings from the upcoming STRASS2 trial pertaining to the impact of neoadjuvant chemotherapy on resected LMS and DD-LPS may better inform our use of neoadjuvant therapy for these histologies.

### 3.4. STRASS2

STRASS2 is currently recruiting and will be the first phase III randomized trial for neoadjuvant systemic therapy in RPS and pelvic sarcomas (Table 1) [71]. It aims to recruit 250 high-risk DD-LPS and LMS patients over 5.5 years and will randomize patients to either upfront surgery or neoadjuvant chemotherapy (Figure 1). It will administer histology-specific anthracycline-based chemotherapy in keeping with results from a multi-institutional study [70] and institutional practices at participating centers: LPS patients will receive three cycles of doxorubicin + ifosfamide, while LMS patients will received three cycles of doxorubicin + dacarbazine.

The histologic inclusion criteria for this study were based on patterns of distant failure observed in DD-LPS and LMS patients at participating centers [19,72]. The goal of this trial is to assess the impact of chemotherapy on disease-specific and overall survival in this population. It also aims to determine the impact of neoadjuvant chemotherapy on the rate and timing of distant failure. It will potentially identify patients who would not have benefitted from surgery—namely, those who progress while on chemotherapy. Novel aspects of this trial that will be especially beneficial are the histologic-specific analyses and correlative studies that will comparatively evaluate specimens for pathologic response to therapy.

### 3.5. Targeted Therapy

As with other rare diseases, clinically actionable molecular targets in sarcoma have the potential to change the landscape of the disease. This is perhaps most evident in gastrointestinal stromal tumors (GIST), for which targeted therapy in the form of imatinib has made sweeping changes to our treatment paradigm, including in the neoadjuvant setting. With respect to retroperitoneal tumors, studies identifying unique gene expression profiles of common RPS histotypes such as LPS have been encouraging but have not yet been translated into therapeutic targets. The ultra-rare mesenchymal tumors for which molecular targets are being clinically investigated (such as PEComas) are unfortunately uncommon in the retroperitoneum.

Pazopanib, a multiple TKI that targets VEGFR and PDGFR, is approved as a second-line agent for advanced STS but is not considered to be an appropriate neoadjuvant agent for any of the common RPS histologies. Among all-comers with STS, it has shown the greatest efficacy in LMS and solitary fibrous tumors (SFT), among others. In the initial phase II study of pazopanib for STS, all but 26% of patients with adipocytic tumors/LPS had experienced tumor progression at 12 weeks, and therefore the LPS stratum was closed [73]. However, synovial sarcomas, LMS, and other sarcoma groups had improved outcomes with this agent, and the phase III trial excluding LPS showed that pazopanib improved PFS in these patients by 3 months [74]. While it is not a substitute for anthracycline-based therapy, pazopanib may provide an alternative for LMS, SFT, or synovial sarcoma patients who have contraindications to doxorubicin/epirubicin and would be offered systemic therapy in the preoperative setting for locally advanced tumors that are not amenable to upfront resection.

### 3.6. Immunotherapy

There is currently no established role for immunotherapy in the neoadjuvant setting for RPS. One phase II trial explored the benefit of the anti-PD-1 therapy pembrolizumab in patients with metastatic STS treated with multiple previous lines of therapy (SARC028) [75]. They demonstrated a 40% objective response (OR) in undifferentiated pleomorphic sarcoma (UPS), a 20% OR in LPS, and no response in LMS [75]. Alliance A091401 studied nivolumab alone or with ipilimumab in a similar population and saw even lower response rates (5% and 16%, respectively), with only two LMS and two UPS patients in the combination group responding [76]. Despite this study being completed in a metastatic patient population, LPS, UPS, and LMS are the most common RPS histologies. Studying the response to immunotherapy in these histologies may yield information that could be applied in the neoadjuvant setting as part of a multimodality approach.

As would be expected, immunobiology across STS subtypes varies dramatically. Though sarcomas were previously considered immunologically inert (‘cold’), we are now discovering that certain subtypes are consistently immunologically responsive (‘hot’) and may be effectively targeted with immunotherapy. Mechanisms accounting for this were explored in a study evaluating the tumor microenvironment (TME) in classically ‘cold’ rhabdomyosarcoma (RMS) and ‘hot’ UPS, as well as LMS [77]. This study showed that, despite a similar tumor-associated macrophage (TAM)-predominant microenvironment across histotypes (corroborated by published RNA gene expression datasets [78,79]), UPS had a higher immune fraction of M2 macrophages and greater PD-L1 expression than RMS [77]. There was also diffuse CD3+ and CD8+ T-cell infiltration as well as diffuse PD-L1 expression identified in UPS tumors compared with RMS, where the majority of T cells (and PD-L1 expression) were concentrated in tertiary lymphoid structures near vascular beds, with low absolute immune cell fractions found elsewhere in the tumor [77]. PD-L1 expression was also greater in UPS than LMS, although not significant. Similar results were found in SARC028’s correlative analyses as well as other studies [80,81]. High PD-1 expression has also been observed in a substantial proportion of resected LPS specimens (65%) in a recent study on the TME of WD- and DD-LPS [82]. Here, intratumoral tertiary lymphoid structures were identified in half of the LPS specimens and had a distribution similar to RMS as described above [82].

PD-L1 expression levels have been correlated with anti-PD-1 treatment response in other cancers and will be an important pillar of translational research as we pursue effective applications of immunotherapy in specific sarcoma subtypes [83,84,85,86]. While none of this has translated into sufficiently strong clinical evidence to be considered part of a neoadjuvant paradigm, histology-specific translational research will inform the design of early phase trials, which should subsequently provide better biomarkers and evidence of pathologic effect in sensitive subtypes.

### 3.7. Active Trials

A French phase II trial (TORNADO) with planned enrollment starting in 2022 will investigate the effect of neoadjuvant chemotherapy and retifanlimab on histologic response for previously untreated resectable RPS of any histology (NCT04968106) [87]. Retifanlimab is a humanized PD-L1 inhibitor that has orphan drug status for Merkel cell carcinoma and anal carcinoma, and it has been studied in phase III trials for endometrial cancer, non-small cell lung cancer, and phase II/III trials for gastric and esophageal cancer. There are no published studies on retifanlimab in soft tissue sarcoma, and it has not been previously explored for resectable disease; however, it is being studied in STS patients with advanced disease in a number of NIH and industry sponsored trials.

In one active phase II trial exploring neoadjuvant nivolumab +/− ipilimumab for resectable retroperitoneal DD-LPS, patients will receive five weeks of upfront immunotherapy (three doses) prior to surgical resection [88]. The primary outcome is pathologic response. While there is no standard of care arm, evidence of pathologic response on surgical specimens may inform the design of subsequent phase III trials intended to evaluate novel neoadjuvant treatment approaches against the standard of care. Of note, other arms of the same trial will compare these treatment regimens in extremity or trunk UPS in conjunction with neoadjuvant radiation.

Finally, though immunotherapy has changed the face of a number of cancers, the rarity of STS has slowed our entry into this field of investigation. Our increasing understanding of the heterogeneous immune signatures of various STS and growing collaborative tissue repositories will likely inform the design of forthcoming studies, enabling patients with a subset of sarcomas to benefit substantially from immunotherapy.

## 4. Multimodality Neoadjuvant Therapy

Although myxoid LPS is not a retroperitoneal histology, it has demonstrated an impressive response to multimodal therapy in recent early-phase clinical trials. In the setting of an established response to both trabectidin and radiotherapy, myxoid liposarcoma—previously treated with anthracycline-based chemotherapy in keeping with the standard of care for other LPS—was investigated in the TRASTS trial, a study of three cycles of neoadjuvant trabectidin and 45 Gy RT. Phase I of TRASTS demonstrated a good safety profile and anti-tumor activity [89]. In the phase II trial, 91.5% of patients completed their preoperative trabectedin, and all patients completed RT and underwent surgery [90]. Patients demonstrated a dramatic improvement in residual tumor burden at the time of surgery, with over half having less than 10% residual tumor volume. One patient developed a local recurrence and two developed a distant recurrence at two years [90]. While other low-grade LPS have not demonstrated the same impressive response to trabectedin as myxoid LPS, trabectedin in combination with RT for grade I–II retroperitoneal RPS may be justified pending longer-term results from the phase II TRASTS study, STRASS, and other studies exploring LPS response to trabectedin.

With a trend toward improved ARFS in LPS patients receiving neoadjuvant RT in STRASS, and STRASS2 exploring chemotherapy for high-risk LMS and DD-LPS, therapies that serve to enhance the effects of traditional RT or chemotherapy agents will be important to explore. Overexpression of the mouse double minute 2 homolog (MDM2) protein from amplification of the *mdm2* gene is the most commonly recognized abnormality in LPS. MDM2 suppresses downstream *p53* function, and so MDM2 inhibition is only an effective strategy for *p53* wild-type tumors. A recent study also identified an activating *PIK3CA* mutation across several LPS samples [91]. Treatment with a PI-103, a dual phosphatidylinositol 3-kinase (PIK3) and mammalian target of rapamycin (mTOR) inhibitor, reduced AKT phosphorylation and expression of anti-apoptotic proteins, and furthermore, exhibited a significant reduction in growth when combined with either cisplatin or doxorubicin chemotherapy [91].

With respect to targeting MDM2 amplification, experimental agents have shown promise in proof-of-concept studies and early trials. [92,93]. While MDM2 inhibitors may grow to be an essential component of therapy for LPS in themselves, they may also contribute to synergistic treatment with radiotherapy. In an in vitro study of human WD-LPS and DD-LPS cell lines, cells were treated with nutlin-3 (an MDM2 inhibitor), radiation, or both [94]. Two of the DD-LPS cell lines that had MDM2 amplification/TP53 wild-type demonstrated radiosensitization through enhanced senescence of polyploidy cells among other possible mechanisms [94]. With the STRASS results suggesting some benefit for LPS with neoadjuvant RT, the shift of MDM2 inhibitors into sarcoma-focused clinical trials may have a substantial impact on how we deliver neoadjuvant therapy.

The SARC028 findings prompted a phase II study of neoadjuvant pembrolizumab in concert with RT for UPS and LPS of the extremities (SU2C-SARC032), which has now completed accrual [95]. Though this study investigates multimodality therapy and does not include retroperitoneal tumors, it may provide valuable information for radiosensitive histology-specific responses to anti-PD-1 therapy that can be extrapolated to RPS. Similarly, the multi-institutional NEXIS study is recruiting non-RPS patients with high or intermediate-grade sarcoma for a study of durvalumab (PD-L1 inhibitor) and tremelimumab (anti-CTLA4) during neoadjuvant RT and in the adjuvant setting [96]. While this study does not include RPS, it may provide important information surrounding the effects of immunotherapy in concert with RT for STS in general, as well as provoking histology-specific hypotheses about the effect of immunotherapy in different sarcoma subtypes.

As we see long-term outcomes from STRASS, anticipate histology-specific neoadjuvant radiotherapy trials in the future, and interpret data from ongoing early-phase trials of other approaches to RPS, the additive role of immunotherapy for susceptible subtypes in the neoadjuvant setting will be essential to explore.

### Intersection of STRASS1 and STRASS2

The histologic subgroup analyses for STRASS2 will be the most important aspect of this new trial: compelling evidence for a chemotherapy effect in LMS, DD-LPS, or both would have to be demonstrated to prompt a shift in our neoadjuvant paradigm, which—until STRASS2 enrollment—uncommonly included systemic therapy for upfront resectable tumors. If this trial demonstrates a positive result for DD-LPS, then this evidence, in conjunction with the clinically compelling findings in the STRASS and STREXIT data, may open up a new line of investigation for multimodality neoadjuvant treatment in DD-LPS. As has been the case with other solid tumors that have moved toward a total neoadjuvant treatment paradigm, an important line of study will be minimizing delays to surgery after integrating RT and chemotherapy. The results of STRASS2 may also inform the design of new neoadjuvant protocols generated from our growing understanding of novel therapies. Current approaches to neoadjuvant treatment, considering the background evidence supporting these trials, are depicted in Figure 2. 

The rate of progression in STRASS1 was high (16%). However, all but one of these patients either had a complete macroscopic resection despite local tumor progression or else went on to develop distant metastases and would have likely failed in the early post-operative period regardless. In addition to sparing patients an unnecessary operation, similar findings in STRASS2 would have the advantage of testing tumor biology with early first-line chemotherapy and positioning them to receive appropriate subsequent treatment at a time when they have a low metastatic burden.

## 5. Conclusions

Determining the role of neoadjuvant chemo- or radiotherapy in RPS is challenging, especially given the difficulty of achieving sufficient enrollment in clinical trials to generate meaningful conclusions. Recent international collaborations have allowed us to develop valuable prospective registries, analyze large and granular retrospective cohorts, and complete the first successful phase III trial in RPS. While STRASS raised a number of questions, it does provide compelling evidence for histology-specific variations in local control for RPS, with a signal for improved ARFS in low-grade LPS. High-grade RPS with a tendency toward distant failure will be enrolled in STRASS2 to explore the impact of neoadjuvant anthracycline-based chemotherapy in this population. As long-term results from these studies unfold, we can also expect ongoing progress in the fields of radiation, targeted therapy, and immunotherapy to add to our existing body of evidence for various RPS histologies. As these arise, continued collaborative efforts will enable us to progress towards optimizing our treatment approaches for these rare cancers.

## Figures and Tables

**Figure 1 cancers-14-01831-f001:**
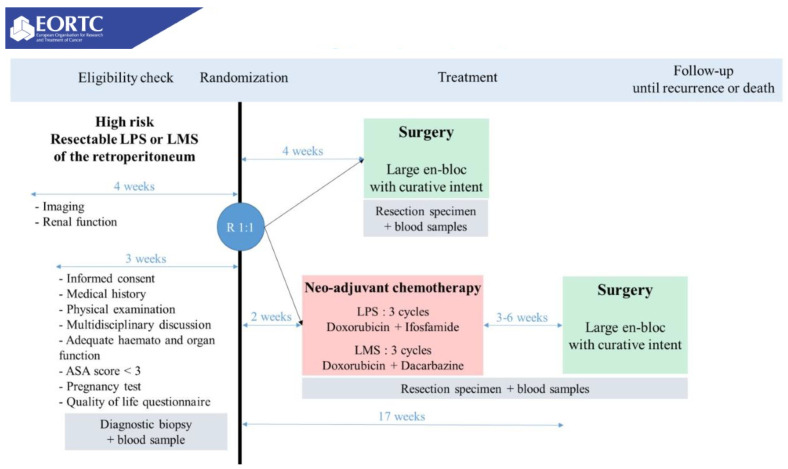
STRASS 2 study design, NCT04031677.

**Figure 2 cancers-14-01831-f002:**
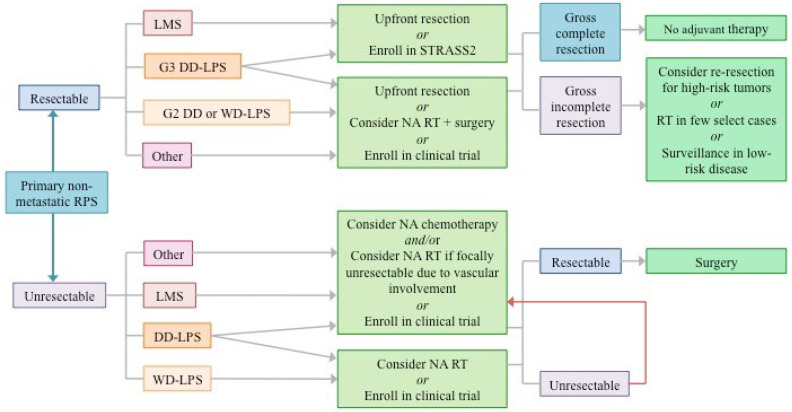
Approaches to neoadjuvant therapy for retroperitoneal sarcomas based on both general and histology-specific results from recent and ongoing studies; consistent with published guidelines from ESMO-EURACAN, TARPSWG, and the NCCN.

**Table 1 cancers-14-01831-t001:** Phase III randomized controlled trials (RCT) evaluating neoadjuvant therapy approaches for retroperitoneal sarcoma (RPS) patients.

Study	Design	N	Population	Intervention and Comparator	Outcomes	Findings
Radiotherapy						
**STRASS**^18^ Bonvalot 2020	Phase III RCT (1:1)	266	Resectable primary RPS	**I**: Neoadjuvant 3DCRT or IMRT (50.4 Gy in 28 fx of 1.8 Gy) + surgery **C**: surgery alone (en-bloc curative intent resection)	Primary: AFRS Secondary: tumor response, DMFS, ARFI, OS, safety, QoL	No difference in ARFS on ITT analysis 3-year ARFS 66% v. 59% on 1st sensitivity analysis ** 3-year ARFS 72% v. 60% on 2nd sensitivity analysis ** In LPS patients, failure reported for 40% receiving RT v. 60% surgery alone.
Systemic therapy						
**STRASS2** NCT04031677	Phase III RCT (1:1)	250 *	Resectable high-risk primary retroperitoneal LMS and LPS	**I**: 3 cycles neoadjuvant chemotherapy (LPS: ADM + ifosfamide, LMS: ADM+DTIC) **C**: surgery alone (en-bloc curative intent resection)	Primary: DFS Secondary: OS, LRFS, RFS, DMFS	Study in progress

*** planned per sample size calculation. ** 1st sensitivity analysis conducted such that local progression on radiotherapy was not regarded as a primary endpoint event for those who had macroscopically complete resection; 2nd sensitivity analysis conducted such that neither local progression nor becoming medically unfit on radiotherapy were regarded as primary endpoint events for those who had macroscopically complete resection. 3DCRT—3D conformal radiotherapy, IMRT—intensity-modulated radiotherapy, ARFS—abdominal recurrence-free survival, DMFS—distant metastasis-free survival, ARFI—abdominal recurrence-free interval, OS—overall survival, QoL—quality of life, ITT—intention to treat, LMS—leiomyosarcoma, LPS—liposarcoma, ADM—doxorubicin, DTIC—dacarbazine, DFS—disease-free survival, LRFS—local recurrence-free survival.

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
