# Peer review of "Neoadjuvant Therapy for Primary Resectable Retroperitoneal Sarcomas—Looking Forward"

_cancers, 2022, doi:10.3390/cancers14071831_

Round 1
Reviewer 1 Report
Reviewer comments:
Comments to the Author
This review summarizes the current evidence related to neoadjuvant chemo- and radiotherapy for retroperitoneal sarcomas, active trials of interest, and future directions in this field. This review is not clear for most of the part and not comprehensive under the topics. Content is very vague. No tables and diagrams were provided to support the readability. Language is not clear throughout the article. Please undergo a thorough check of the manuscript for typographical and grammatical errors.
Author Response
Thank you for your comments. Your recommendations for tables and diagrams to make the review more visually accessible have been noted and incorporated. We appreciate this feedback, and similar recommendations were made by other reviewers.
Any identified errors have been corrected. More specific comments as to relevant content that the authors may have missed are welcome.
Reviewer 2 Report
This article is a descriptive review of preoperative treatment for resectable retroperitoneal sarcoma. It is well described to the point and is of high clinical value. On the other hand, the content is somewhat complex and difficult to understand, and the following improvements should be considered.
1. An addition of a summary table for pivotal trials in accordance with the PICO format
2. An addition of a summary table on options of preoperative treatment of resectable retroperitoneal sarcoma
The above revisions will help the reader understand the content of this article more visually.
Author Response
Thank you very much for the helpful feedback. Schemas of the STRASS trials have been added to the manuscript. A summary of neoadjuvant approaches to RPS have been depicted as well.
Reviewer 3 Report
The manuscript reviewed retroperitoneal STS. The highlight of this review should be STRASS and STRASS2. I recommend the authors add treatment schema of these studies.
Minor comment
Why the regimen of chemotherapy in patients with LMS and DDLPS is different?
Author Response
Thank you very much for the helpful feedback. Schemas of both of these trials have been added to the manuscript. The chemotherapy regimens for STRASS 2 differ for LMS and DDLPS based on previous studies as well as institutional protocols and preferences at the enrolling institutions. While there is no trial-level data for this decision, neither of these regimens has been demonstrated to be inferior, and aligning with current institutional practices will improve accrual.
Reviewer 4 Report
Overall, I think this is an outstanding review well organized and documented, however there are some issues need to be addressed.
I strongly recommend to include a table or may be two with a summarized of all clinical trial (chemotherapy, radiotherapy or both), the readers would be benefit to find quickly any trial.
Please define all the acronyms that appear in the manuscript, there are several whose meaning we do not know.
Sarcomas with NTRAK alterations are emerging entities, I wonder whether there is any trial involving NTRAK sarcomas??, I cannot find any on this manuscript.
Would be interesting to include a figure with inclusion criteria, treatment plan , follow up and goals of STRASS 2 trial, it is not clear on the manuscript.
Author Response
Thank you very much for the helpful feedback. Schemas of the STRASS trials have been added to the manuscript. A summary of common neoadjuvant approaches to RPS has also been presented.
The manuscript has been reviewed for acronyms that we failed to define initially. Thank you for pointing this out.
Finally, while we certainly agree that NTRK fusion sarcomas are a growing topic of interest and study, they tend to occur in infantile fibrosarcomas predominantly and occasionally in synovial or other sarcomas that are notably rare in the retroperitoneum. Therefore, because the scope of this review was limited to retroperitoneal tumors, we only incorporated studies for treatment approaches that would be applied to common retroperitoneal sarcomas.
Round 2
Reviewer 1 Report
Authors have incorporated the suggestions and the review looks better than before.